DOI: 10.1038/s41467-017-02770-z | OPEN

# Molecular basis for the specific and multivariant recognitions of RNA substrates by human hnRNP A2/B1

Baixing Wu[1], Shichen Su[1], Deepak P. Patil[2], Hehua Liu[1,3], Jianhua Gan[3], Samie R. Jaffrey [2] & Jinbiao Ma [1]

Human hnRNP A2/B1 is an RNA-binding protein that plays important roles in many biological processes, including maturation, transport, and metabolism of mRNA, and gene regulation of long noncoding RNAs. hnRNP A2/B1 was reported to control the microRNAs sorting to exosomes and promote primary microRNA processing as a potential m$^6$A "reader." hnRNP A2/B1 contains two RNA recognition motifs that provide sequence-specific recognition of RNA substrates. Here, we determine crystal structures of tandem RRM domains of hnRNP A2/B1 in complex with various RNA substrates, elucidating specific recognitions of AGG and UAG motifs by RRM1 and RRM2 domains, respectively. Further structural and biochemical results demonstrate multivariant binding modes for sequence-diversified RNA substrates, supporting a RNA matchmaker mechanism in hnRNP A2/B1 function. Moreover, our studies in combination with bioinformatic analysis suggest that hnRNP A2/B1 may mediate effects of m$^6$A through a "m$^6$A switch" mechanism, instead of acting as a direct "reader" of m$^6$A modification.

[1] State Key Laboratory of Genetic Engineering, Collaborative Innovation Centre of Genetics and Development, Department of Biochemistry, Institute of Plant Biology, School of Life Sciences, Fudan University, Shanghai 200438, China. [2] Department of Pharmacology, Weill Cornell Medicine, Cornell University, New York, NY 10065, USA. [3] State Key Laboratory of Genetic Engineering, Collaborative Innovation Center of Genetics and Development, Department of Physiology and Biophysics, School of Life Sciences, Fudan University, Shanghai, 200438, China. Baixing Wu and Shichen Su contributed equally to this work. Correspondence and requests for materials should be addressed to J.M. (email: majb@fudan.edu.cn)

Heterogeneous nuclear ribonucleoproteins (hnRNPs) play a variety of roles in regulating transcriptional and post-transcriptional gene expression, including RNA splicing, polyadenylation, capping, modification, export, localization, translation, and turnover[1, 2]. Each hnRNP contains at least one RNA-binding domain (RBD), such as RNA recognition motif (RRM), K-Homology (KH) domain, or an arginine/glycine-rich box[3]. Sequence-specific association between hnRNPs and their RNA targets are typically mediated by one or more RBDs, which usually bind short, single-stranded RNA[4, 5], but in some instances, also recognize structured RNAs[6].

As a core component of the hnRNP complex in mammalian cells, hnRNP A2/B1 is an abundant protein and has been implicated in numerous biological processes. The *HNRNPA2B1* gene encodes two protein isoforms, A2 and B1, through alternative splicing. The B1 isoform contains an insertion of 12 amino acids (aa) at its N terminus[7, 8]. Both isoforms have an RNA-binding domain (RBD) composed of tandem RRMs separated by a 15-aa linker, and a C-terminal Gly-rich low complexity (LC) region that includes a prion-like domain (PrLD), an RGG box, and a PY-motif containing a M9 nuclear localization signal (PY-NLS)[9–11]. These domains are represented schematically with residue numbers based on the hnRNP B1 isoform (Fig. 1a).

hnRNP A2/B1 is linked to several biological processes and diseases, especially neurodegenerative disorders, e.g., mutations in core PrLD in hnRNP A2/B1 cause multisystem proteinopathy and amyotrophic lateral sclerosis (ALS), through promoting excess incorporation of hnRNP A2/B1 into stress granules and driving the formation of cytoplasmic inclusions in animal models[9]. hnRNP A2/B1 also regulates hESC self-renewal and pluripotency[12].

hnRNP A2/B1 has multiple effects on RNA processing through binding specific sequence. It can bind to HIV-1 RNA, causing nuclear retention of the vRNA, as well as microRNAs, sorting them into the exosomes through binding "EXO-motifs"[13, 14]. A transcriptome-wide analysis of hnRNP A2/B1 targets in the nervous system identified a clear preference for UAG(G/A) motifs confirmed by three independent and complementary in vitro and in vivo approaches[15, 16]. This is consistent with previous studies indicating that hnRNP A2/B1 binds specifically to UAGGG, GGUAGUAG, or AGGAUAGA sequences[17, 18]. Another recent study demonstrated that hnRNP A2/B1 recognizes a consensus motif containing UAASUUAU (S = G or C) in the 3′ UTR of many mRNAs and helps recruiting the CCR4–NOT deadenylase complex[19]. In addition to participating the regulation of mRNAs, hnRNP A2/B1 is also involved in the activities of many other RNA species. For example, hnRNP A2/B1 can promote association of the long noncoding RNA HOTAIR with the nascent transcripts of HOTAIR target genes, thus to mediate HOTAIR-dependent heterochromatin initiation[20].

Recently, hnRNP A2/B1 was proposed to bind RNA transcripts containing $N^6$-methyladenosine, a widespread nucleotide modification in mRNAs and noncoding RNAs[21, 22]. hnRNP A2/B1 was found to mediate m[6]A-dependent nuclear RNA processing events by binding G(m[6]A)C-containing nuclear RNAs in vivo and in vitro, in which hnRNP A2/B1 associates with a subset of primary microRNA transcripts through binding m[6]A, promoting primary microRNAs processing by recruiting the microprocessor complex Drosha and DGCR8[23].

Although a series of studies with different approaches pointed out the multitudinous functions of hnRNP A2/B1 mediating by diverse RNA motifs in vivo, no detailed mechanism for the binding specificities has been determined. Thus, more biochemistry and structural studies are essential to understand the RNA-binding properties of hnRNP A2/B1 at the molecular level. Here, we report the crystal structures of hnRNP A2/B1 in complex with variant RNA targets to unveil the RNA-binding specificities and multivariant characteristics. Moreover, our structural data along with RNA-binding and bioinformatic analysis do not support that hnRNP A2/B1 functions as an m[6]A "reader," since no significant preference for m[6]A modification by either tandem RRMs or full-length protein of hnRNP A2/B1 are observed.

## RESULTS

**Crystal structure of hnRNP A2/B1 bound to an 8mer RNA.** To elucidate the RNA-binding properties of tandem RRMs of hnRNP A1/B1, we purified a number of truncations of the hnRNP A2/B1 protein. Using isothermal titration calorimetry (ITC) method, we characterized the RNA-binding activities of each construct with a set of RNA oligonucleotides of sequences according to previously determined binding motifs. ITC results showed that the construct containing the N-terminal fragment (aa 1–11) and two RRM domains, i.e., aa 1–195, can bind target RNAs with high affinity. In addition, deletion of the N-terminal fragment had no obvious impact on the RNA binding. This suggested that the construct containing the tandem RRMs of hnRNP A2/B1 is sufficient for binding target RNAs.

We first determined the crystal structure of RRMs (aa 12–195) of hnRNP A2/B1 bound to the 8-nt RNA oligonucleotide 5′-$A_1G_2G_3A_4C_5U_6G_7C_8$-3′ (termed 8mer RNA), which is derived from a recent individual-nucleotide-resolution CLIP study[16, 23]. ITC analysis showed that binding of the 8mer RNA occurs at a 1:1 ratio with a $K_d$ of 276.2 nM (Fig. 1b). The crystal structure of RRMs (aa 12–195) in complex with the 8mer RNA molecule was determined to 2.60 Å resolutions, details about data collection and structure refinement are summarized in Supplementary Table 1. The tandem RRMs and an RNA molecule are in one asymmetric unit (Fig. 1c); both RRM domains of hnRNP A2/B1 adopt the characteristic RRM fold, which is a typical β1α1β2β3α2β4 topology consisting of an antiparallel four-stranded sheet adjacent to two helices on the opposite side, similar to previously determined RRM structures of other RNA-binding proteins using both crystallographic and NMR methods[24] (Fig. 1c). Each RNA molecule is bound by an RRM1 domain from one hnRNP A2/B1 molecule in an asymmetric unit and an RRM2 domain from another hnRNP A2/B1 molecule in adjacent asymmetric unit (Fig. 1d).

**RRM1 specifically recognizes AGG motif.** The AGG motif of 8mer RNA substrate is specifically recognized mainly by RRM1 (Fig. 1e). For the recognition of the adenine at the first position ($A_1$), the 2′-OH group forms a hydrogen bond and π–π interactions with the side chain of His108. Besides hydrogen bonding interactions, base stacking with the Phe24 on the other side also contributes to the definition of the binding environment (Fig. 1f, g). The 2′-OH of $G_2$ forms a hydrogen bond with the side chain of Arg99, and $N^1$ groups of $G_2$ form hydrogen bonds with the carboxyl group of the main chain of Val97 while $N^7$ interacts with the side chain of Lys22. The base of this guanine $G_2$ engages in stacking interactions with the base of the benzene ring of Phe66 and guanidyl group of Arg99 (Fig. 1f, g). $N^1$ and $N^2$ of $G_3$, the last nucleotide in the core recognition AGG motif, are hydrogen bonded to the side chain of Asp49, and $O^6$ and $N^7$ are recognized by the side chain of Arg99 (Fig. 1f, g). However, the RNA substrate from $A_4$ to $C_8$ are not well specifically recognized (Fig. 1f). The $N^6$ of $A_4$ is hydrogen bonded to the main chain of Lys186, and the 2′-OH group is hydrogen bonded to the side chain of Glu192 (Fig. 1f, g). The base of $U_6$ is sandwiched between Phe115 and $U_4$ base via π–π stacking, whereas the $O^4$ of $U_6$ forms hydrogen bonds with the amino group of the main chain of Arg185 (Fig. 1f, g).

**Both RRMs are involved in recognition of the 10mer RNA**. The crystal structure of hnRNP A2/B1(aa 12–195) in complex with the 8mer RNA did not provide insight into specific RNA recognition by RRM2. We thus designed another RNA oligonucleotide shown in Supplementary Table 2, based on the speculation that

RRM2 might recognize UAG according to previous sequencing results[17], [18]. This RNA contains both the AGG motif and the UAG motif. ITC results confirmed that the 10-nt RNA oligo 5′-$A_0A_1G_2G_3A_4C_5U_6A_7G_8C_9$-3′ (termed 10mer RNA) has a higher affinity (Fig. 1b and Supplementary Table 2). We successfully

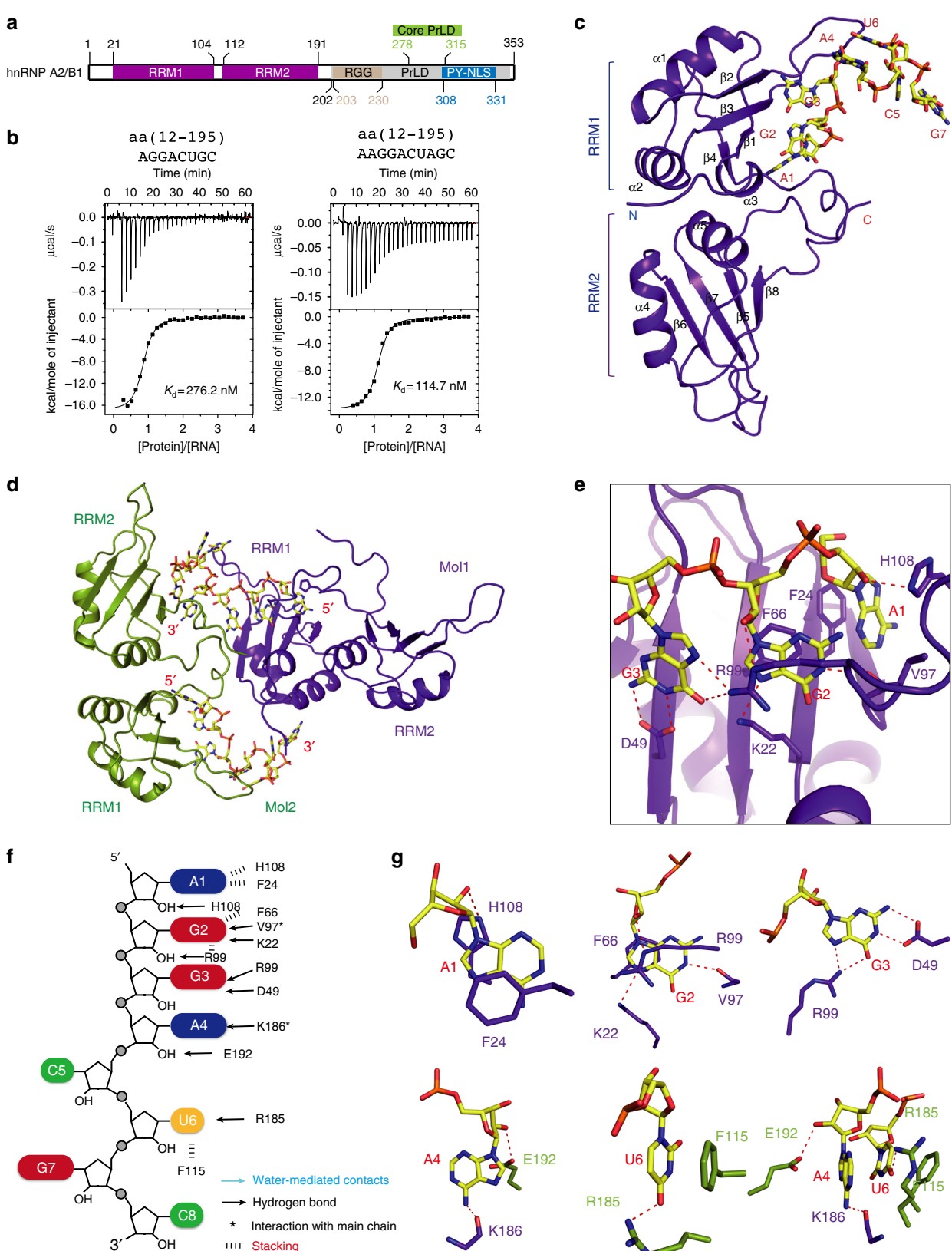

obtained the crystal of hnRNP A2/B1(12–195) in complex with the 10mer RNA, which was determined to 1.85 Å resolution (Supplementary Table 1). Similar to the previous complex structure with 8mer RNA, there is also only one protein molecule and one RNA molecule in the asymmetric unit. However, this time the RNA molecule is recognized by both RRM1 and RRM2 from two hnRNP A2/B1 molecules, of which the other one is from the asymmetric molecule (Fig. 2a). The 10mer RNA molecule adopts a single-stranded conformation accommodated into a positively charged groove comprised by the canonical RNA-binding surface of the RRM1 and RRM2 from two hnRNP A2/B1 proteins, in a 5′–3′ direction from RRM1 to RRM2 (Fig. 2b). Unlike the complex structure of 8mer RNA in which only RRM1 is involved in specific recognition, the complex structure containing 10mer RNA also shows specific recognition by RRM2 (Fig. 2c). When the structures of RRM1 bound to 8mer and 10mer complex are superimposed, the AGG motifs of the two RNA substrates also superimpose well (Fig. 2d). However, the conformation of the rest of the oligos are dramatically different. Notably, the 10mer RNA is more stretched than the 8mer RNA (Fig. 2d). Moreover, the root-mean-square deviation of protein backbones between the two structures is merely 0.4 Å, suggesting the protein has not changed substantially when binding to the two different RNA targets.

**The RRM2 specifically recognizes UAG motif.** Due to the higher resolution, more detailed interactions are observed in the complex structure of 10mer RNA than in the 8mer RNA complex. Though the recognition of the AGG motif in the two structures is quite similar, more specific recognitions of $A_1$ and $G_2$ in the 10mer RNA complex structure are observed. $N^1$ of $A_1$ is recognized by the main chain amine group of Val97, whereas $N^6$ and $N^7$ form hydrogen bonds with the Lys94 side chain either directly or mediated by a water molecule (Fig. 3a, b). $G_2$ has the most complicated interacting network in this structure, in which two more $G_2$ base-specific recognitions are seen in the 10mer complex. These are the side chains of Gln19 and Ser102, which cooperatively bond to $O^6$ and $N^2$ of $G_2$, respectively (Fig. 3b). In addition to specific recognition of the AGG motif, $N^6$ of the 5′-end extended adenine ($A_0$) is hydrogen bonded to the Glu92, and the $N^1$ and $N^6$ of $A_4$ form hydrogen bonds with the side chain of Lys120 and Asn181 mediated by water, respectively (Fig. 3a, b).

Unlike the 8mer RNA complex structure, the UAG motif in the 10mer RNA is specifically recognized by RRM2. The side chain of Arg185 hydrogen bonds to both the 2′-OH and $O^2$ group of $U_6$, and the $N^3$ and $O^4$ groups are hydrogen bonded to the side chain of Glu183 and Arg99, respectively (Fig. 3a, b). The base of $A_7$ is also clamped by Phe115 and Met193 through hydrophobic interactions; $N^7$ of $A_7$ forms hydrogen bond with Arg185; $N^1$, $N^6$, and the phosphate group are hydrogen bonded to the main chain of Leu188, Lys186, and the side chain of Arg185, respectively (Fig. 3a, b). In addition to the hydrogen bond formed between phosphate group of $G_8$ and the side chain of Arg153, the $O^6$ of $G_8$

base is recognized by the side chain of Lys113, and the base of $G_8$ has another π–π stacking with Phe157 (Fig. 3a, b). The complete structure of $C_5$ and $C_9$ cannot be seen in our structure.

The chains of RRM1 (12–110) and RRM2 (111–195) can be superimposed with a root-mean-square deviation of 0.901 Å with a high sequence identity (Supplementary Fig. 1a). A superimposition of RRM1-AAGG with RRM2-ACUAGC indicated that the recognition of the AG core motif is very similar in RRM1 and RRM2 (Supplementary Fig. 1b, c). We thereafter mutated residues involved in specific recognition of AGG by RRM1 and UAG by RRM2, both of which reduced binding affinities according to ITC (Fig. 3c and Supplementary Fig. 2). Although the results of these amino acid mutations lined with expectations, the nucleotide mutations of 10mer RNA, especially the UAG nucleotides recognized by RRM2, have only moderate effects on the binding affinities (Fig. 3d, Supplementary Fig. 3, and Supplementary Table 2).

**Multivariant RNA recognition modes of RRM1 and RRM2.** In order to understand the molecular basis for hnRNP A2/B1 recognizing different RNA sequences, we grew the crystals of hnRNP A2/B1 (aa 12–195) in complex with different 10mer RNA mutants. Three complex structures containing RNA mutants A1G, U6G, and A7U were determined at high resolution (Fig. 4a, b, c). For the A1G mutant (5′-$A_0G_1G_2G_3A_4C_5U_6A_7G_8C_9$-3′), the AGG motif is shifted to 5′-end and recognized by RRM1 in a manner almost identical to the wild-type 10mer RNA structure (Fig. 4d). In addition, the $G_3$ is recognized by the side chains of Arg99 and Glu18 of RRM1 through hydrogen bond formation (Fig. 4d, e, f). Therefore, RRM1 of hnRNP A2/B1 can specifically recognize an AGGG motif as demonstrated in the A1G mutant structure. Additionally, RRM2 recognizes UAG in the same manner as was seen in the structure of wild-type 10mer RNA (Fig. 4d, e).

In contrast, hnRNP A2/B1 adopts distinct strategies for binding another two RNAs that contained mutations in the UAG motif recognized by RRM2. When $U_6$ is substituted with G in the U6G complex (5′-$A_0A_1G_2G_3A_4C_5G_6A_7G_8C_9$-3′), $G_6$ lost two hydrogen bonds from Arg99 and Arg185, only keeping hydrogen bonds with Glu182. However, the recognition of the AG core motif by RRM2 is still well maintained (Fig. 4g, h, i). Interestingly, the recognition of AAGG by RRM1 is exactly same as AGGG in the A1G RNA mutant, though their binding modes of AG motif are different, suggesting that RRM1 can accommodate various purine-rich sequences (Fig. 4g, h, i and Supplementary Fig. 4a, b). For the A7U RNA mutant (5′-$A_0A_1G_2G_3A_4C_5U_6U_7G_8C_9$-3′), $U_7$ forms hydrogen bonds with the side chain of Lys113 and Glu142 and forms a π–π stacking interaction with Phe157. More interestingly, $U_6$ adopts a sandwich-like interaction mode with Phe115 and the $A_4$ base, which is exactly same as the $U_6$ in the 8mer RNA substrate (Fig. 5j, k, l and Supplementary Fig. 4c, d). This suggests that RRM2 can accommodate the pyrimidine-rich UU sequence.

**Fig. 1** Overview of the structure and ITC of hnRNP A2/B1 in complex with 8mer RNA. **a** Schematic representation of the domain architecture of hnRNP A2/B1. RRM: RNA recognition motif, PrLD: prion-like domain, NLS: nuclear location signal, RGG: arginine-glycine-glycine box. **b** ITC results of hnRNP A2/B1 (12–195) with 8mer and 10mer RNA targets. Solid lines indicate nonlinear least-squares fit the titration curve, with $\Delta H$ (binding enthalpy kcal mol$^{-1}$), Ka (association constant), and N (number of binding sites per monomer) as variable parameters. Calculation values for $K_d$ (dissociation constant) are indicated. **c** Cartoon representation of RRMs in complex with 8mer RNA. The RNA backbone is colored in yellow shown by stick. The RRMs are colored in purple-blue. **d** Molecules from two adjacent asymmetric units. The molecule from another asymmetric unit is colored in green. **e** Intermolecular contacts between RRM1 and 8mer RNA from $A_1$ to $G_3$. All dashed lines in this study indicate distance <3.2 Å. **f** Schematic showing RRMs interactions with 8mer RNA 5′-AGGACUGC-3′. **g** Close-up view showing the specific recognition of $A_1$, $G_2$, $G_3$, $A_4$, and $U_6$

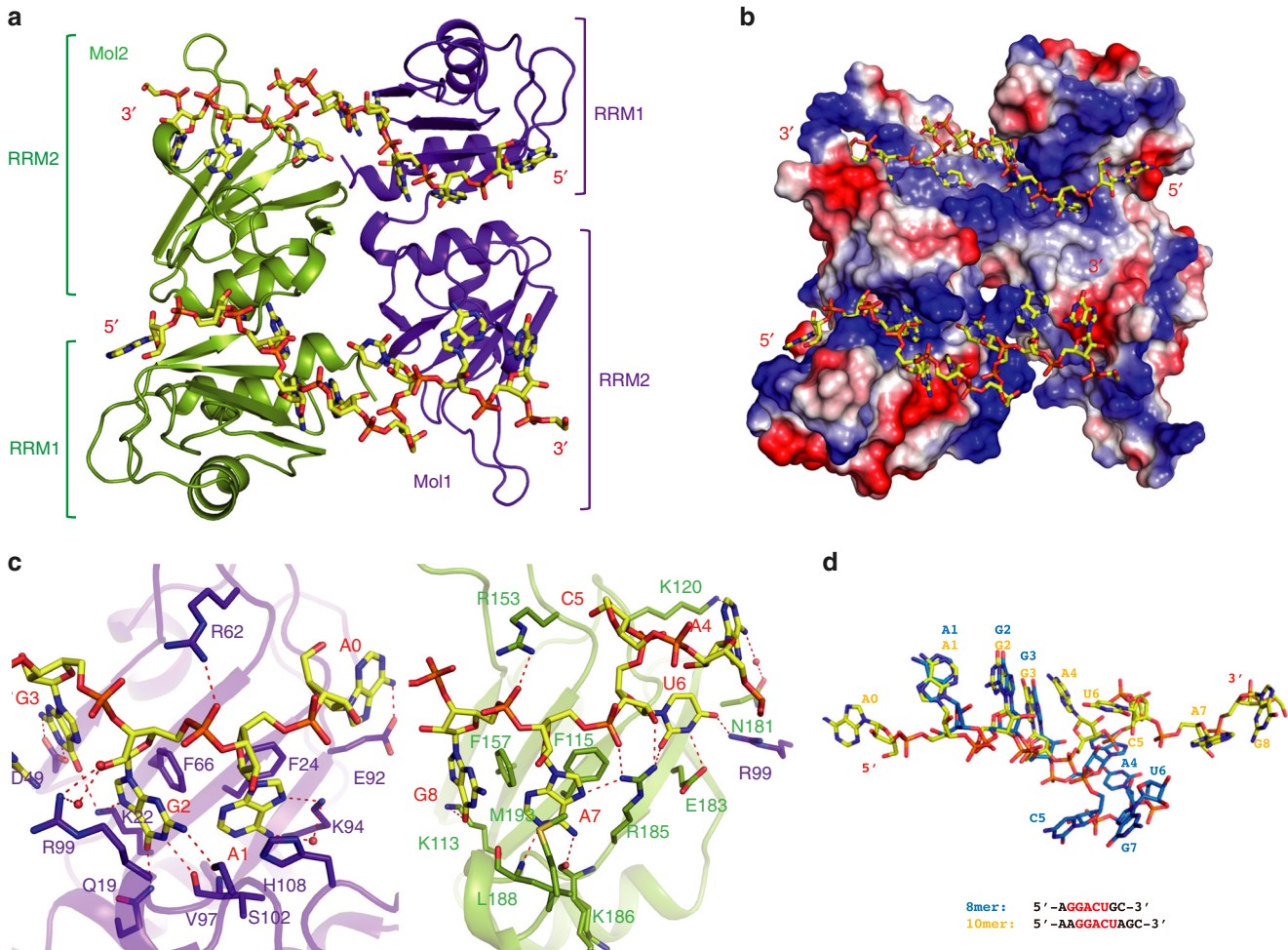

**Fig. 2** Overview of RRMs in complex with 10mer RNA. **a** Cartoon representation of RRMs in complex with 10mer RNA 5′-AAGGACUAGC-3′. The RNA backbone is colored in yellow shown by stick. The molecule from the adjacent asymmetric unit is colored in green. **b** Surface representation of RRMs−10mer complex. **c** Intermolecular contacts between RRM1 and residues of 10mer RNA 5′-AAGG-3′, and RRM2 with residues of RNA 5′-ACUAGC-3′. **d** Superposition of 8mer and 10mer RNA substrates. The 8mer RNA 5′-AGGACUGC-3′ is colored in marine and the 10mer RNA 5′-AAGGACUAGC-3′ is colored in yellow

Meanwhile, the AAGG recognition by RRM1 in A7U RNA mutant is very similar to 8mer and 10mer RNAs, but different from A1G and U6G RNA mutant.

Unlike the effect of mutating the UAG motif recognized by RRM2, which just slightly reduced binding affinities, mutation of the AGG motif recognized by RRM1, such as G2C and G3C RNA mutants, have more obvious effects (Fig. 3d). Although we did not obtain crystal structures of these two mutants, our biochemical and structural studies suggested that RRM1 has more stringent recognition for purine-rich AG motif containing RNA sequences, but RRM2 seems to have more broad compatibility to recognize different RNA sequences, including canonical UAG motif, purine-rich GAG, and pyrimidine-rich UU sequences.

**hnRNP A2/B1 binds two antiparallel RNA strands.** A superimposition of all five structures obtained in this study suggested that hnRNP A2/B1 binds two antiparallel RNA strands using RRM1 and RRM2 concurrently (Fig. 5a). The two RRM domains in hnRNP A2/B1, similar to hnRNP A1 both in crystal structure and in solution[25–27], are held together in a fixed geometry without flexibility (Supplementary Fig. 5a). It is notable that there are extensive interactions between RRM1 and RRM2 from the same hnRNP A2/B1 molecule, including three salt-bridge

interactions of Asp76-Lys168, Arg95-Asp164, Arg82-Asp162, and hydrophobic interactions between Phe20 with Leu171 (Fig. 5b), which are also observed in hnRNP A1 (Supplementary Fig. 5b). In addition, the last β-strand in RRM1 and the first β-strand in RRM2 have the same orientation, which forces the two RRM domains to bind two RNA strands antiparallelly, because the linker between the two domains blocks the binding of RNA targets from the same strand (Fig. 5c, d). In contrast, RNA substrates in most known structures in complex with tandem RRM proteins are bound as single-stranded RNAs and their orientations are from RRM2 to RRM1, such as HuD, HuR, PABP, U2AF65, and TDP-43 (Fig. 5e, f, g, h)[28–32].

**hnRNP A2/B1 does not specifically recognize m6A-modified RNA.** In order to assess the hypothesis that hnRNP A2/B1 might be a direct m6A "reader" as proposed in a previous study[23], the m6A motif GGACU is included in the 8mer and 10mer RNAs. As shown in the crystal structures, there is no obvious aromatic cage-like surface that can potentially bind the m6A nucleotide (Fig. 6a, b), which was shown to be the key m6A-specificity element in previous structural studies of YTHDF1, YTHDC1, and MRB1 complexed with GGm6ACU (Fig. 6c and Supplementary Fig. 6)[33]. The crystal structure of hnRNP A2/B1 in complex with GGm6ACU could not be obtained. However, we were able to

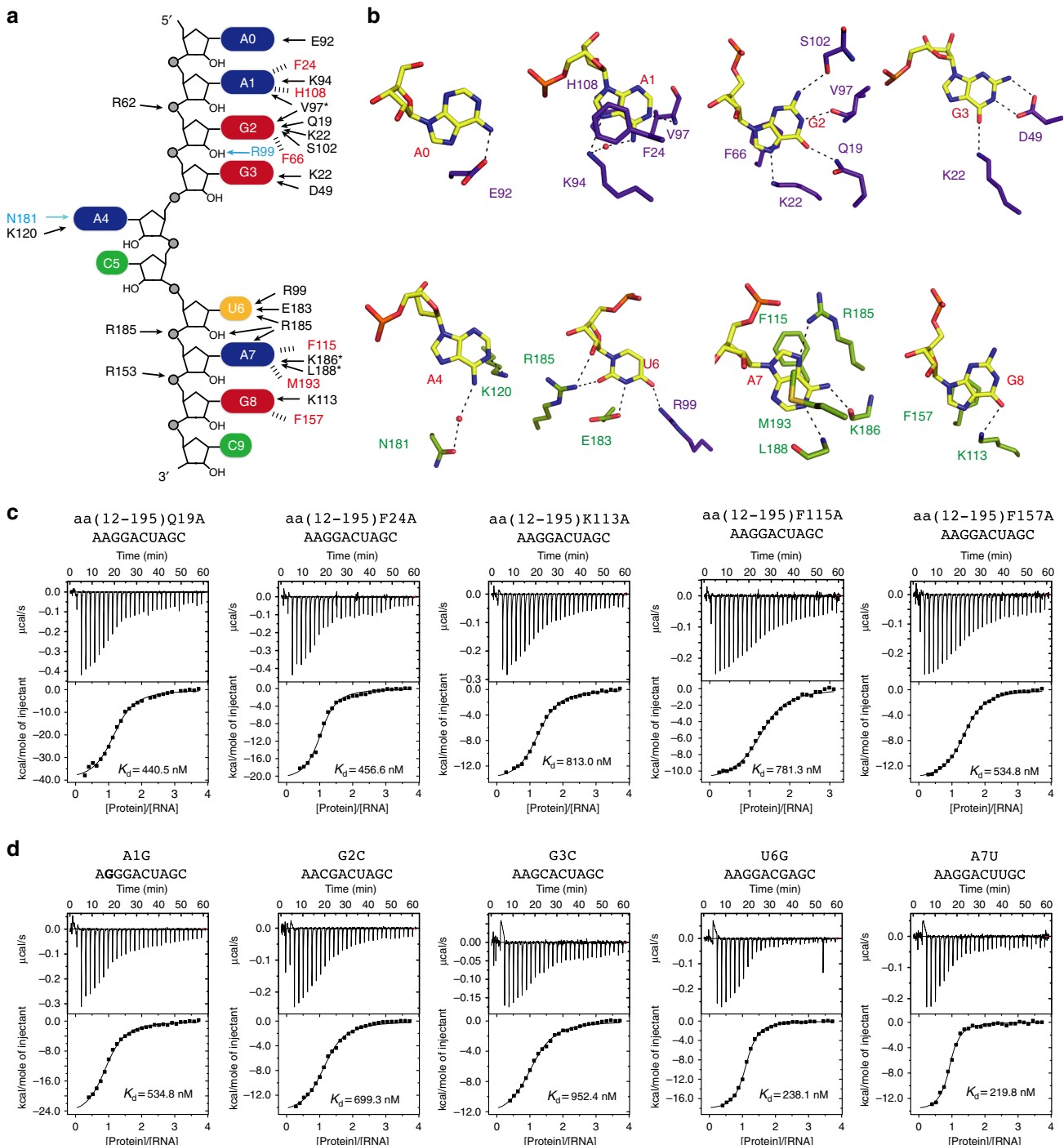

**Fig. 3** Detailed interactions between RRMs and 10mer RNA. **a** Schematic showing RRMs interactions with 10mer RNA sequence. **b** Close-up view showing the specific recognition from $A_1$ to $G_8$. **c** Mutagenesis study by ITC experiments between protein mutants and 10mer RNA substrate. **d** ITC experiments between hnRNP A2/B1 RRMs and RNA mutants A1G, G2C, G3C, U6G, and A7U

detect binding of hnRNP A2/B1 with the 8mer and 10mer RNA in which an adenosine was replaced with an m6A. In both cases, the m6A is present within its preferred GGACU sequence context. Notably, the ITC results indicated that the binding affinities of the m6A-containing 8mer RNA and 10mer RNA to the tandem RRM (12–195) were reduced onefold and tenfold, respectively, compared to the non-methylated RNA (Fig. 6d).

The N6 atoms of $A_4$ in both complex structures of 8mer and 10mer RNAs form hydrogen bonds with hnRNP A2/B1 directly or through a water molecule, which may provide a possible

reason why m6A modification would reduce the binding affinity (Figs. 1g and 3b). However, these data do not exclude the possibility that full-length hnRNP A2/B1 may form a m6A-binding cage through its C-terminal fragment, which contains a RGG box region (Fig. 1a). This domain may also contribute in some way to RNA binding, but was not included in our structural study (Supplementary Fig. 7a). Therefore, we purified the full-length hnRNP A2/B1 with a His6-tag or without the tag and a construct containing RGG box (residue 1–249), and used them to measure the binding affinities with various RNA substrates by

EMSA and ITC experiment (Supplementary Fig. 7b, c, d). The EMSA analysis indicated that full-length hnRNP A2/B1 has a slightly weaker binding affinity to the RNA with m⁶A modification than the one without m⁶A (Fig. 6e and Supplementary

Fig. 7e). The ITC results using the full-length hnRNP A2/B1 or RGG box containing construct (residue 1–249) also showed similar trend as the ITC results using tandem RRMs. These ITC data, along with the EMSA results, suggest that full-length

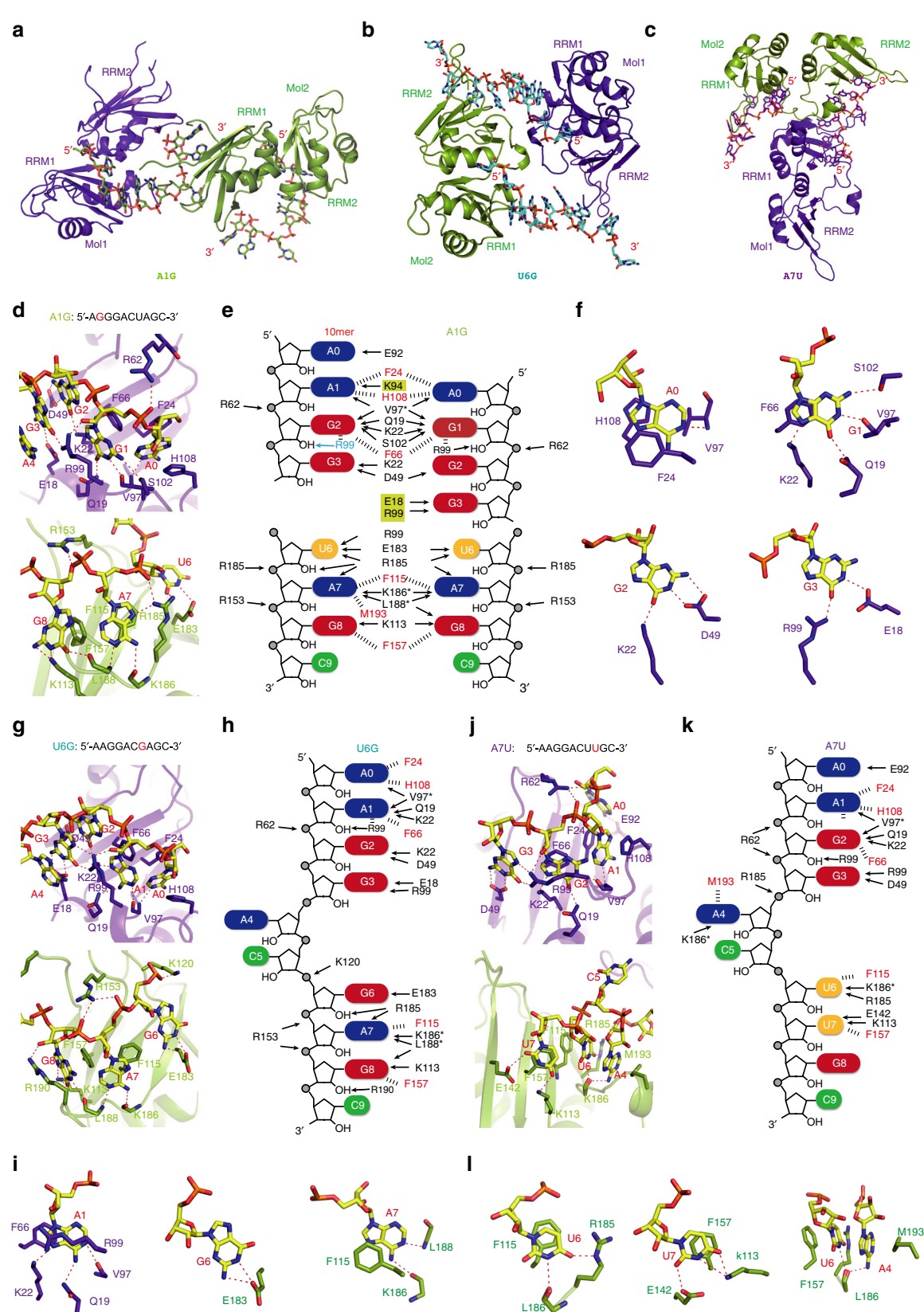

hnRNP A2/B1 does not specifically recognize or show enhanced binding to these m$^6$A-modified RNA substrates in vitro (Supplementary Fig. 7f, g).

**m$^6$A sites bound by hnRNP A2/B1 in vivo**. To better understand the potential binding interactions of m$^6$A and hnRNP A2/B1 in cells, we examined the in vivo binding properties of hnRNP A2/B1. For this analysis, we used a set of 186 nuclear m$^6$A sites comprising m$^6$A sites in *XIST*, *NEAT1*, and *MALAT1*, which have been mapped at single-nucleotide resolution using miCLIP (m$^6$A individual-nucleotide-resolution crosslinking and immunoprecipitation)[34]. In this analysis, we quantified binding at each of these 168 mapped m$^6$A residues by assigning each m$^6$A residue an "intensity value", which was the normalized number of miCLIP reads that overlapped each m$^6$A residue[34]. The intensity value is influenced by transcript abundance and m$^6$A stoichiometry. We next determined the binding of hnRNP A2/B1 at each of these m$^6$A sites based on the normalized number of mapped hnRNP A2/B1 HITS-CLIP tags at the m$^6$A site. For most m$^6$A residues, there was no correlation between m$^6$A intensity and hnRNP A2/B1 binding, although 12 m$^6$A sites showed proximal hnRNP A2/B1 binding, which might be the result of coincidental proximity between m$^6$A and a non-methylated consensus site recognized by hnRNP A2/B1. As a control, we analyzed YTHDC1, a nuclear YTH domain-containing m$^6$A reader[35, 36]. A similar analysis of YTHDC1 showed increasing YTHDC1 binding with increased m$^6$A levels for essentially all m$^6$A sites (Fig. 6f). Thus, unlike hnRNP A2/B1, YTHDC1 appears to function as a general nuclear m$^6$A reader.

Our finding that only a small subset of nuclear m$^6$A sites are positioned near hnRNP A2/B1, and therefore could potentially be involved in a direct m$^6$A-hnRNP A2/B1 interaction, is compatible with the previous analysis by Alarcon et al.[23]. Alarcon et al. reported that only 17% of their total m$^6$A-seq clusters overlap with the hnRNP A2/B1 tag clusters. To determine if YTHDC1 shows greater overlap with m$^6$A than hnRNP A2/B1 does, we performed a similar cluster overlap analysis. Approximately 43% of the miCLIP clusters from total RNA and 56% clusters from miCLIP of poly(A) RNA (Fig. 6g) showed an overlap with YTHDC1 clusters ($P < 0.0001$). Thus, YTHDC1 has a considerably higher overlap with m$^6$A than hnRNP A2/B1. Therefore, this analysis again supports the idea that YTHDC1 is the predominant nuclear m$^6$A reader compared to hnRNP A2/B1.

## Discussion

The RNA-binding domain of hnRNP A2/B1 comprises two RNA recognition motifs, RRM1 and RRM2, which is followed by a C-terminal glycine-rich region. hnRNP A2/B1 was previously demonstrated to bind UUAGGG and UAG RNA motifs through various analyses[37, 38]. Recent CLIP-Seq data further showed that hnRNP A2/B1 prefers to bind A/G-rich sequences[17]. However, there was no molecular basis for the recognition of different RNA substrates of hnRNP A2/B1. Here, we determined the crystal structures of the tandem RRMs of hnRNP A2/B1 in complex with various RNA substrates, revealing the molecular details of specific target RNA recognition and shedding light on the mechanism for hnRNP A2/B1 in various RNA-mediated biological functions.

The specific recognition of the AG core motif by both RRM1 and RRM2 is highly consistent with previous studies showing that hnRNP A2/B1 can bind the A2RE sequence[39] and provides an explanation for how sumoylated hnRNP A2/B1 directs the loading of specific EXO-miRNAs into exosomes by binding GAGG, the so-called EXO motif[14]. Furthermore, our results provide the structural basis for hnRNP A2/B1 binding to the UA-rich UAASUUAU motif in the 3′ UTR of some mRNAs, which was shown to be necessary for loading the CCR4-NOT complex to mRNAs[19]. Taken together, our structures illustrate the sequence-specific RNA-binding properties of hnRNP A2/B1 and give support to previously reported diverse binding sites[17, 18] (Supplementary Table 3).

The two RRM domains in hnRNP A2/B1 interact with each other extensively in a fixed antiparallel orientation, and the binding surfaces of RRM1 and RRM2 align in the same plane, which forces hnRNP A2/B1 to bind two antiparallel RNA strands or a single-stranded RNA with a long connecting loop. This binding property of hnRNP A2/B1 offers a molecular basis for the previously described "matchmaking" hnRNP A2/B1-HOTAIR interaction, which requires multiple nucleotide recognition motifs within HOTAIR[20]. hnRNP A2/B1 shares similar antiparallel arrangements of its bound RNA as the polypyrimidine tract-specific splicing regulator PTB, in which the two RRM domains interact each other extensively and bind two antiparallel RNA strands. In the PTB–RNA complex, RRM3 and RRM4 form a heterodimer mediated by a hydrophobic interface, and bring together two remote RNA pyrimidine tracts[40]. Moreover, two hnRNP A2/B1 proteins bound to same RNA strands can adopt various orientations, as seen in the structures of the 10mer RNA and the two 10mer RNA mutants A1G and U6G (Fig. 5j, k, l and Supplementary Fig. 5c, d). These diverse orientations are mainly due to the absence of direct interactions between the RRM domains bound to the same RNA strands. This feature may also be involved in RNA-templated aggregation and the formation of hnRNP A2/B1-containing protein–RNA granules in vivo[16].

It has recently been demonstrated that hnRNP A2/B1 specifically recognizes m$^6$A-modified RNAs[23]. These RNAs share the m$^6$A consensus sequence RGm$^6$ACH and directly bind to the m$^6$A mark with high affinity in vivo and in vitro[23]. Prior to this study, the YTH domain was shown to be a "reader" of m$^6$A. However, in addition to directly binding-specific proteins, m$^6$A can affect RNA binding through an indirect mechanism. This has been shown with two proteins, HuR and hnRNP C, both of which contain RRM domains and do not directly bind m$^6$A. In the case of hnRNP C, m$^6$A facilitates hnRNP C binding to a UUUUU-tract in mRNAs and long noncoding RNAs (lncRNAs) by promoting local unfolding of RNA. This unfolding is due to the weaker base pairing of U with m$^6$A compared to A[41]. m$^6$A induced RNA unfolding and increases the accessibility of hnRNP

**Fig. 4** RNA mutants indicate mutivariant-binding mode of hnRNP A2/B1 RRMs. **a** Structure of RRMs (12–195) in complex with A1G-RNA 5′-AGGGACUAGC-3′. **b** Structure of RRMs (12–195) in complex with U6G RNA 5′-AAGGACGAGC-3′. **c** Structure of RRMs (12–195) in complex with A7U-RNA 5′-AAGGACUUGC-3′. **d** Intermolecular contacts between RNA and RRMs in A1G complex. RRM1 is colored in purple-blue, RRM2 is colored in green. **e** Schematic representation of the comparison of different intermolecular interactions between 10mer RNA 5′-AAGGACUAGC-3′ and A1G-RNA 5′-AGGGACUAGC-3′. **f** Close-up view showing the specific recognition from A$_1$ to G$_3$. **g** Intermolecular contacts between RNA and RRMs in U6G complex. **h** Schematic representation of intermolecular interactions in the U6G complex. **i** Close-up view showing the specific recognition of A$_1$, G$_6$, and A$_7$ in U6G complex. **j** Intermolecular contacts between RNA and RRMs in A7U complex. **k** Schematic representation of intermolecular interactions in A7U complex. **l** Close-up view showing the specific recognition of U$_6$, U$_7$, and the stacking interactions involved in A$_4$, U$_6$, F157, and M193 in A7U complex

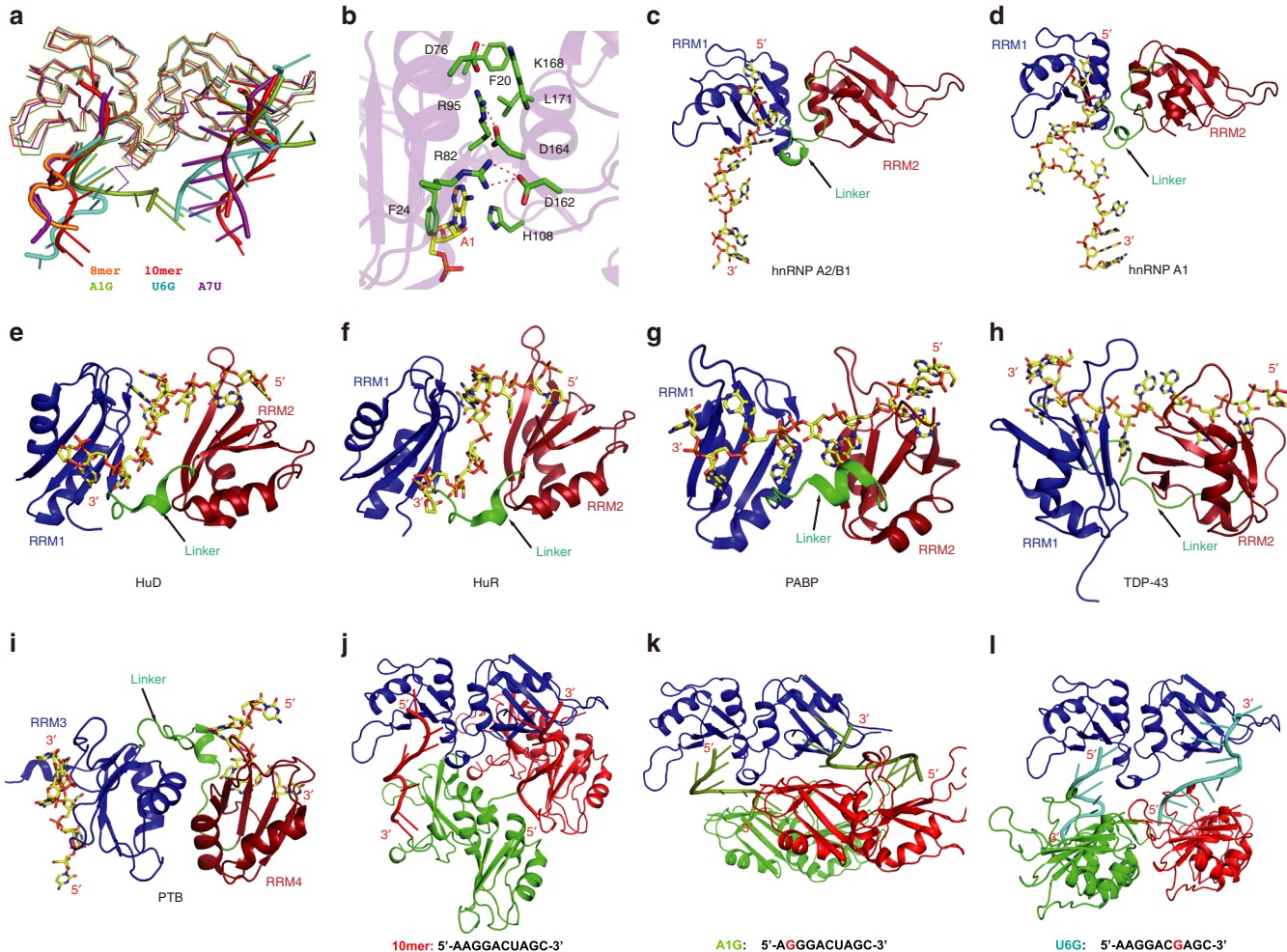

**Fig. 5** The RNA binding by RRMs adopts an antiparallel mode. **a** Superimposition of different structure complex. 8mer-RNA: 5′-AGGACUGC-3′ is colored in orange, 10mer-RNA: 5′-AAGGACUAGC-3′ is colored in red, A1G-RNA: 5′-AGGGACUAGC-3′ is colored in green, U6G: RNA 5′-AAGGACGAGC-3′ and A7U-RNA: 5′-AAGGACUUGC-3′ are colored in cyan and purple, respectively. **b** Interactions between RRM1 and RRM2; the amino acids participating in interactions are colored in green and RNA is colored in yellow. **c** The overall structure of hnRNP A2/B1-10mer RNA complex. **d** hnRNP A1 in complex with DNA. **e–h** The overall structure of HuD-RNA, HuR-RNA, PABP-RNA, and TDP-43-RNA. From **c** to **h**, RRM1 is colored in blue and RRM2 is colored in red, the linker is colored in green pointed out with a black arrow, the RNA backbone is colored in yellow shown by stick. **i** The overall structure of PTB–RNA complex with an antiparallel RNA-binding mode. RRM3 is colored in blue and RRM4 is colored in red, other labels are the same as from **c** to **h**. **j–l** Crystal packing interactions in 10mer, A1G and U6G. To illustrate the detailed packing interactions of hnRNP A2/B1 carrying two antiparallel RNA stands with other hnRNP A2/B1 molecules, three hnRNP A2/B1 molecules and two RNA strands of each complex are selected to show

C to single-stranded RNA, and is therefore termed an "m6A-switch"[42]. HuR, also known as ELAVL1, has been found to preferentially bind to the 3′-UTR region of mRNAs that lack m6A[43]. In this case, m6A could impede the formation of a structured RNA motif needed for HuR binding. Our structural study, combined with biochemistry and bioinformatic results, suggest that m6A switches may account for the previously seen enhanced hnRNP A2/B1 binding adjacent to m6A. Instead of direct binding to m6A, m6A may promote accessibility of hnRNP A2/B1 to certain binding sites, thereby explaining how m6A can facilitate the ability of hnRNP A2/B1 to enhance nuclear events such as pri-miRNA processing. Further in vitro and in vivo investigations will be required to uncover the details of this mechanism.

## Methods

**Preparation of protein samples.** Plasmids encoding different fragments of hnRNP A2/B1 were PCR amplified from the human cDNA. PCR products were double digested with restriction endonuclease *BamHI* and *XhoI*, then ligated into a modified pET-28a plasmid carrying the Ulp1 cleavage site. Mutations were generated based on the overlap PCR. Recombinant plasmids were confirmed by DNA

sequencing and transformed into *Escherichia coli* BL21 (DE3) to produce target proteins with N-terminal hexahistidine-sumo fusions. *E. coli* cells were cultured in LB medium at 37 °C with 50 mg/l kanamycin until the $OD_{600}$ reached 0.6–0.8, then the bacteria were induced with 0.2 mM isopropyl-β-D-thiogalactoside (IPTG) at 18 °C for 16 h. Bacteria were collected by centrifugation, resuspended in buffer containing 20 mM Tris-HCl pH 8.0, 500 mM NaCl, 20 mM imidazole pH 8.0, and lysed by high pressure. Cell extracts were centrifuged at $38,758 \times g$ for 1 h at 4 °C. Supernatants were purified with Ni-NTA (GE), the target protein was washed with lysis buffer and then eluted with a buffer containing 20 mM Tris-HCl, pH 8.0, 500 mM NaCl, and 500 mM imidazole. Ulp1 protease was added to remove the N-terminal tag and fusion protein of the recombinant protein and dialyzed with lysis buffer 3 h. The mixture was applied to another Ni-NTA resin to remove the protease and uncleaved proteins. Eluted proteins were concentrated by centrifugal ultrafiltration, loaded onto a pre-equilibrated HiLoad 16/60 Superdex 75-pg column in an Äkta-purifier (GE Healthcare), eluted at a flow rate of 1 ml/min with the same buffer containing 10 mM Tris-HCl pH 8.0, 100 mM NaCl. Peak fractions were analyzed by SDS-PAGE (15%, w/v) and stained with Coomassie brilliant blue R-250. Purified fractions were pooled together and concentrated by centrifugal ultrafiltration. The concentration was determined by $A_{280}$. The protein was concentrated to 10 mg/ml for crystallization trials.

**RNA oligonucleotides.** The RNA oligonucleotides (8mer-m6A: 5′-AGGm6A-CUGC-3′, 10mer-m6A: 5′-AAGGm6ACUAGC-3′) with m6A modification were ordered from Dharmacon (Thermo Scientific.), and the other unmodified or

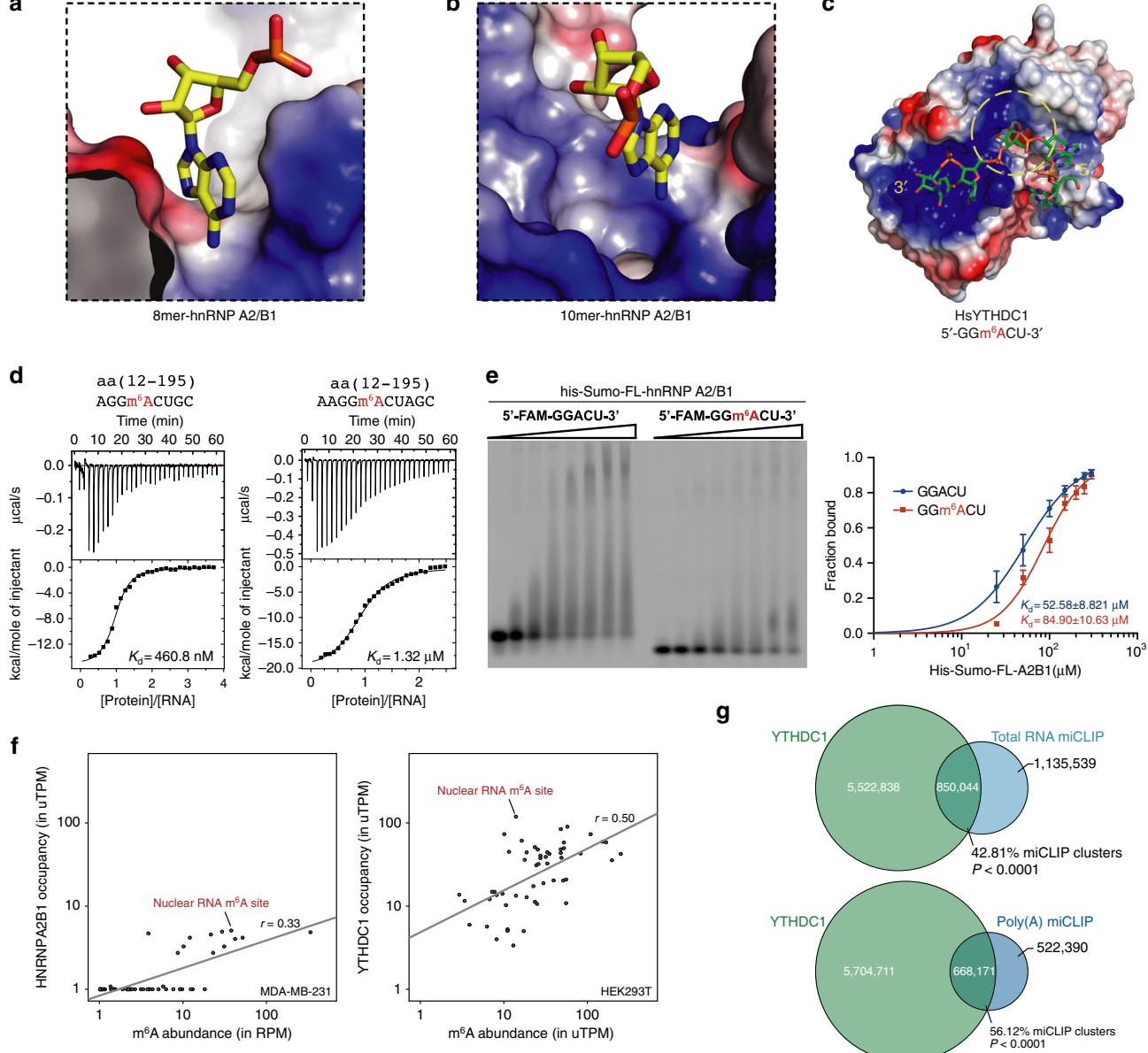

**Fig. 6** hnRNP A2/B1 does not specifically recognize m6A-modified RNA. **a** Surface representation of the environment around $A_4$ in 8mer RNA complex. **b** Surface representation of the environment around $A_4$ in 10mer RNA complex. **c** Surface representation of the canonical N6-methylated adenosine binding mode in HsYTHDC1. The aromatic cage is circled with yellow dashline. **d** ITC data of hnRNP A2/B1(12–195) with 8mer and 10mer RNA targets carried N6-methylated adenosine. **e** EMSA experiment shows the binding affinity of full-length hnRNP A2/B1 with 5′-FAM-labeled RNA substrates with or without m6A modification. Uncropped gel image is shown in Supplementary Fig. 7e. The data represent the mean of three independent experiments, with standard deviation (SD) values indicated by error bars. **f** YTHDC1 shows preferential binding to m6A sites in nuclear RNA compared to hnRNP A2/B1. For hnRNP A2/B1, the m6A-Seq reads that overlapped with each m6A site was plotted on the x-axis, and the HITS-CLIP reads that overlap with each site were plotted on the y-axis. A similar analysis was used to examine YTHDC1 binding at these m6A sites. miCLIP reads that overlapped with the m6A sites were plotted on the x-axis, and the YTHDC1 iCLIP reads that overlapped with the m6A sites were plotted on the y-axis. **g** YTHDC1-m6A tag cluster overlap. A Venn diagram indicating the cluster overlap is shown. Roughly, 43 and 56% of miCLIP tag clusters from total cellular RNA and poly(A) RNA showed a significant overlap with the YTHDC1 iCLIP clusters, respectively

unlabeled RNA oligonucleotides were synthesized by the IDT-394 synthesizer in our own lab. The 5′-FAM-labeled RNA chains (5′-FAM-GGACU-3′ and 5′-FAM-GGm6ACU-3′) were ordered from Bioneer Corporation. All the RNA oligonucleotides used for crystallization and biochemical experiments in this study are summarized in Supplementary Table 2.

**Crystallization and data collection.** hnRNP A2/B1 RRMs (12–195) in complex with 8-nt-RNA 5′-AGGACUGC-3′ was crystallized using the hanging drop vapor diffusion method by mixing 1 μl of protein–RNA mixture (molar ratio 1:1.2) and 1 μl of reservoir solution at 20 °C. The crystals suitable for X-ray diffraction were grown in reservoir solution consisting of 0.1 M Tris pH 8.5 and 25% (w/v)

polyethylene glycol 3,350 (Hampton Research). hnRNP A2/B1 RRMs (12–195) in complex with 10-nt-RNA 5′-AAGGACUAGC-3′ was screened as above. The crystal suitable for X-ray diffraction was grown in reservoir solution consisting of 0.2 M Tri-sodium citrate and 20% (w/v) polyethylene glycol 3,350 (Hampton Research). A1G, U6G, and A7U were crystallized as the methods mentioned above in solution containing 20% PEG 3,000, 0.1 M sodium citrate pH 5.5; 20% PEG 3,350, 0.2 M lithium sulfate, 0.1 M Bis–Tris pH 6.5; 25% PEG 1,500, 0.1 M MMT pH 9.0, respectively. Data collection were performed at 100 K with cryo-protectant solution (reservoir solution supplemented with an additional 20% (v/v) glycerol). Diffraction data were collected using a wavelength of 0.97776 Å at beamline BL18U1 of the Shanghai Synchrotron Radiation Facility (SSRF).

**Structure determination and refinement**. For hnRNP A2/B1 RRMs (12–195)-8-nt complex, the diffraction data set was processed and scaled using HKL3000. The phase was determined by molecular replacement using the program Phaser with the structure of UP1 (PDB code: 1U1Q) as the search model[44]. Cycles of refinement and model building were carried out using REFMAC5 and COOT until the crystallography $R_{factor}$ and $R_{free}$ converged to 19.16% and 23.62%, respectively[45, 46]. Ramachandran analysis showed that 96.0 of the residues were in the most favored region, with 4.0% in the additionally allowed region. For hnRNP A2/B1-10-nt complex, the diffraction data set was processed and scaled using the HKL3000 package. The phase was determined by molecular replacement using the program Phaser with the hnRNP A2/B1(12–195) model collected before as the search model. Cycles of refinement and model building were carried out using REFMAC5 and COOT until the crystallography $R_{factor}$ and free $R_{free}$ converged to 18.39% and 22.27%, respectively. Ramachandran analysis showed that similarly to hnRNP A2/B1-8-nt, 99% of the residues were in the most favored region, with 1% in the additionally allowed region. For another three complex structures, the same methods were used to solve the structures as mentioned above. The details of data collection and processing are presented in Supplementary Table 1. All structure figures were prepared with PyMOL (DeLano Scientific).

**ITC measurements**. ITC assays were carried out on a MicroCal ITC200 calorimeter (Malvern) at 25 °C. The buffer used for proteins and RNA oligomers was 10 mM HEPES pH 8.0, 50 mM KCl, 1 mM EDTA, and 1 mM BME. The concentrations of proteins were determined spectrophotometrically. The RNA oligonucleotides were diluted in the buffer to 5–15 µM. The ITC experiments involved 20–30 injections of protein into RNA. The sample cell was loaded with 250 µl of RNA at 5 µM and the syringe with 80 µl of protein at 100 µM; for weak complexes, the measurement was repeated with increased concentrations. Reference measurements were carried out to compensate for the heat of dilution of the proteins. Curve fitting to a single binding site model was performed by the ITC data analysis module of Origin 7.0 (MicroCal) provided by the manufacturer. $\Delta G^o$ of protein–RNA binding was computed as $RT\ln(1/K_D)$, where $R$, $T$, and $K_D$ are the gas constant, temperature and dissociation constant, respectively.

**EMSA**. Aliquot of 0.5 µM of FAM-labeled RNA was mixed with increasing concentrations of full-length hnRNP A2/B1 proteins in a buffer containing 10 mM HEPES pH 8.0, 50 mM KCl, 1 mM EDTA, and 5 mM beta-mercaptoethanol in a total volume of 10 µl and incubated at room temperature for 30 min. The electrophoresis was performed with 6% native-PAGE at 4 °C in running buffer containing 0.5× Tris-borate-EDTA (TBE) buffer. The gel was visualized by using a Typhoon FLA-9000 (GE Healthcare) using a method for FAM (Laser 488 nm). Bound and free RNA were quantified using ImageJ. Binding curves were fit individually using GraphPad Prism 6.0 software fitting with "One site − Specific binding with Hill slope" (GraphPad Software). Curves were normalized as percentage of bound oligonucleotides and reported is the mean ± SD of the interpolated $K_d$ from three independent experiments.

**Analytical gel filtration**. Proteins of RRM1 and RRM2 domains of hnRNP A2/B1 were purified using the same procedure as hnRNP A2/B1 (12–195). Analytical gel filtration chromatography was performed at 4 °C using a Superdex 75 10/300 global column (GE Healthcare) pre-equilibrated in 20 mM Tris, pH 8.0, 100 mM NaCl. Aliquots of 100 µl of samples including various hnRNP A2/B1 constructs were injected at a flow rate of 0.3 ml/min. To study complex formation, RRM1 and RRM2 proteins were mixed and incubated on ice for at least 1 h prior to loading.

**Next-generation sequencing data analysis**. Nuclear hnRNP A2/B1HITS-CLIP sequence data were obtained from a previously published study[23] (GEO accession number: GSE70061, SRA accession numbers: SRR2071655 and SRR2071656, last update date: 21 Jun 2015). In addition to the raw data, the author uploaded sequence alignment files, GSM1716539_A2B1_HITS_CLIP_1.bedgraph.gz and GSM1716539_A2B1_HITS_CLIP_2.bedgraph.gz were also obtained from the GEO database for comparison purposes. Robust crosslinking-induced mutation sites (CIMS) (FDR ≤0.001) in hnRNP A2/B1 HITS-CLIP data were called using a method published elsewhere[47]. UV-induced deletion sites[48] were used as hnRNP A2/B1-binding sites.

Nuclear m6A-seq data from MDA-MB-231 cells were obtained from a previously published study[49] (GEO accession number: GSE60213, SRA accession numbers: SRR1539129 and SRR1539130, last update date: 15 Nov 2016). Adapter-free, high-quality sequence reads were aligned to the hg19 genome build using bowtie2 according to the source publication[49]. RPM (reads per million mapped reads) was calculated using bedtools.

For YTHDC1 binding at m6A sites in HEK293T cells, miCLIP sequencing data[34] (GEO accession number: GSE63753) and YTHDC1 iCLIP data[35] (GEO accession number: GSE78030) were obtained from the GEO database. Sequence alignments were carried out according to the respective publications. Images of genome alignments were prepared using IGV genome browser and Adobe Illustrator.

**Comparison of hnRNPA2B1 and YTHDC1 binding at m6A sites**. hnRNP A2/B1 or YTHDC1 binding and m6A stoichiometry at 10 bp flanking miCLIP sites[34] on nuclear RNAs such as *MALAT1*, *NEAT1*, and *XIST* was compared using an XY-scatter plot in R. Only m6A sites conforming a non-BCANN consensus were considered for this analysis. These represent unique sites obtained from merging (mergeBed -s -d 2) of CIMS- and CITS-based m6A site calls from ref. [34]. All the rRNA, tRNA, and mitochondrial genomic miCLIP sites were removed. Tag counting was performed using the bedtools suite. Tag counts (uTPM + 1) were compared using scatter plots and Pearson correlation coefficients (r) were determined in R. Cluster overlap analysis was carried out using bedtools intersect tool (intersectBed -s -u).

**Data availability**. The coordinates that support the findings of this study have been deposited in the Protein Data Bank with accession codes 5EN1 for RRMs-8mer-RNA complex and 5HO4 for RRMs−10mer RNA structures. A1G, U6G, and A7U are 5WWE, 5WWF, and 5WWG, respectively. Other data in this study are available from the corresponding author on reasonable request.

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

## Acknowledgements

We thank the staff from BL18U1 beamline of the National Facility for Protein Science in Shanghai (NFPS) at the Shanghai Synchrotron Radiation Facility for assistance during data collection. We thank Dr Jinzhong Lin (Fudan University) for helpful discussions and critical manuscript reading. This work was supported by grants from the National Natural Science Foundation of China (31230041) and the National Basic Research Program of China (2011CB966304 and 2012CB910502) to J.M., and the National Institutes of Health (R01 CA186702) to S.R.J.

## Author contributions

B.W. and S.S. expressed, purified, and grew crystals of the hnRNP A2B1–RNA complex and performed the biochemical assays. B.W., S.S., and H.L. synthesized the RNA oligonucleotides. B.W., J.M., and J.G. collected X-ray diffraction data and solved the human hnRNP A2B1–RNA complex structure. D.P. carried out the bioinformatic analysis. B.W., J.M., D.P., and S.J. wrote and revised the manuscript. J.M. supervised the structural and biochemical study. S.J. supervised the bioinformatic study.

## Additional information

**Competing interests:** The authors declare no competing financial interests.

