## [Peer Review File · Nature Communications]

Reviewers' comments:

Reviewer #1 (Remarks to the Author):

In this paper, Wu et al. report several crystal structures of the RRM of hnRNPA2B1 bound to different RNA clients. These novel structures provide new insights into how hnRNPA2B1 engages different RNAs in different ways. They then present negative data to show that hnRNPA2B1 does not specifically recognize m6A modified RNA. Overall, the new structures are exciting and help us understand the complexity of how hnRNPA2B1 engages diverse RNAs and how it differs from other hnRNPs. In my view, the work is of interest but is not ready for publication until two points are addressed:

1. Several domains are misassigned in Fig. 1. In hnRNPA2, the prion-like domain is actually found from residues 190-341 (see Kim et al. Nature. 2013), whereas the authors have just depicted the 'core prion-like domain'. Moreover, hnRNPA2 has a PY-NLS in the prion-like domain (residues 296-319, see Lee et al. Cell. 2006. 126(3):543-58.), which controls nuclear localization and not the NLS proposed by the authors in the N-terminal region (which has not been shown to be an NLS). An RGG box found in the prion-like domain is also not shown. These errors need to be fixed.

2. ITC data showing full-length hnRNPA2B1 binding to various RNAs should be shown to compare to the RRM construct. The prion-like domain (defined above) actually has an RGG box, which is also anticipated to contribute to RNA binding. Hence, it is critical that the data with full-length full-length hnRNPA2B1 binding to various RNAs is shown.

Reviewer #2 (Remarks to the Author):

hnRNP A2/B1 is a RNA-binding protein involved with mRNA splicing, processing, export, and primary microRNA processing. It forms complexes with at least 20 other different hnRNPs and heterogeneous nuclear RNAs in the nucleus. The authors solved high resolution crystal structures of hnRNP A2/B1 with various RNA substrates, illustrating the molecular mechanism underline AGG and UAG motif recognition by A2B1 RRM1 and RRM2 domains. Together with biochemical and bioinformatics analysis, they demonstrate hnRNP A2/B1 works as a RNA matchmaker by binding different RNA substrates, rather than a direct m6A reader.

The crystal structures provide direct and solid evidence for the RNA substrate recognition by hnRNP A2/B1 and the in vitro EMSA and ITC results further confirm the conclusion. With YTHDC1 as a positive control, the authors employ bioinformatic analysis to demonstrate that only limited m6A-seq clusters overlap with the hnRNP A2/B1 tag clusters and there is no correlation between m6A intensity and hnRNP A2/B1 binding, suggesting that hnRNP A2/B1 might play a role in m6A structural switch. The multivalent binding characteristics of the hnRNP A2/B1 protein and the antiparallel orientation of the RNA strands bound by two RRM domains provide the molecular basis for hnRNP A2/B1-mediated RNA granule formation, which is significant insights given importance of this protein. I support publication of this work.

Reviewer #3 (Remarks to the Author):

The manuscript by Wu et al reports a comprehensive structural and biochemical analysis of RNA recognition by hnRNP A2/B1. The author determined the first crystal structures of the tandem RRM domains of hnRNP A2/B1 bound to five different RNAs. These structures combined with binding data reveal the conserved and variable features of RNA recognition for two RRM domains. The

authors show that RRM1 and RRM2 have different specificity for AGG and UAG, respectively, and different stringency in recognition. Although tens of RRM-RNA complex structures are available, the RNA binding specificity of each RRM is determined by intricate and unpredictable interactions and has to be characterized individually. One important feature of these structures is the antiparallel arrangement of two RRM domains, which is different from most other tandem RRM-RNA structures. This arrangement has important consequence in RNA recognition and requires the two target RNAs recognized by RRM1 and RRM2 to be folded back or from different molecules.

hnRNP A2/B1 has been implicated in recognizing m6A modification. The current study provides evidence that hnRNP A2/B1 does not directly recognize m6A, but disfavors m6A, clarifying its role in m6A recognition.

Overall, the study is technically sound and the structural data are of high quality. The study reveals the structural basis of binding specificity of hnRNP A2/B1 and the unusual arrangement of RRM domains and is important for understanding the in vivo binding targets and function of hnRNP A2/B1.

Major concerns.

1. In all the determined structures, the two RRM domains adopt the same unusual antiparallel arrangement. It is probably the reason why the RNA strands in all the crystals are bound to different protein molecules, because the linker between two target sites is too short to allow the RNA adopt a fold-back conformation. The antiparallel arrangement and the interactions between two RRM domains is an important feature of hnRNP A2/B1 structure. My major concern is whether the antiparallel arrangement is caused by crystal packing? Could the authors show whether the RRMs domains are fixed in solution. Is there any interaction between two RRMs domains when they are separated?

Minor concerns

1. The protein-RNA interaction is described in too much detail. As the interactions of RRM1 are similar in the 8-mer and 10-mer complexes, repetitive description should be avoided.

Response to Reviewers' comments:

We appreciate the reviewers for their positive and insightful comments that enabled us to improve the quality of our manuscript. Here below we outline our responses to the points raised by reviewers.

Reviewer #1 (Remarks to the Author):

In this paper, Wu et al. report several crystal structures of the RRM s of hnRNPA2B1 bound to different RNA clients. These novel structures provide new insights into how hnRNPA2B1 engages different RNAs in different ways. They then present negative data to show that hnRNPA2B1 does not specifically recognize m6A modified RNA. Overall, the new structures are exciting and help us understand the complexity of how hnRNPA2B1 engages diverse RNAs and how it differs from other hnRNPs. In my view, the work is of interest but is not ready for publication until two points are addressed:

1) Several domains are misassigned in Fig. 1. In hnRNPA2, the prion-like domain is actually found from residues 190-341 (see Kim et al. Nature. 2013), whereas the authors have just depicted the 'core prion-like domain'. Moreover, hnRNPA2 has a PY-NLS in the prion-like domain (residues 296-319, see Lee et al. Cell. 2006. 126(3):543-58.), which controls nuclear localization and not the NLS proposed by the authors in the N-terminal region (which has not been shown to be an NLS). An RGG box found in the prion-like domain is also not shown. These errors need to be fixed.

Response:

We thank the reviewer for his positive comments on our works. We agreed that there are several errors in Fig. 1. We fixed these errors in the revised Fig. 1a, and corrected related parts in the revised manuscript. The residue numbers for the schematic representation of domains in hnRNP A2/B1 is based on B1 isoform instead of A2 isoform that is lacking 12 aa in N-terminal compared to B1 isoform. The C-terminal low complexity domain (residue 202-353) is assigned as Prion-like domain (PrLD) as the reviewer suggested. The core Prion-like domain (core PrLD) is assigned based on the definition in the reference (Kim et al. Nature. 2013). The PY-motif containing nuclear localization signal with a M9 sequence (PY-NLS) is assigned according to the reference (Lee et al. Cell. 2006). The RGG box is shown in the revised Fig. 1a as in the reference (Harrison & Shorter, Biochem. J., 2017), and is also shown in Supplementary Fig. 7.

Reference:

1. Lee B.J. *et al.* (2006). Rules for nuclear localization sequence recognition by karyopherin beta 2. *Cell* 126: 543-558

2. Kim, H.J. *et al.* (2013) Mutations in prion-like domains in hnRNPA2B1 and hnRNPA1 cause multisystem proteinopathy and ALS. *Nature* 495, 467-73

3.

2) ITC data showing full-length hnRNPA2B1 binding to various RNAs should be shown to compare to the RRM construct. The prion-like domain (defined above) actually has an RGG box, which is also anticipated to contribute to RNA binding. Hence, it is critical that the data with full-length hnRNPA2B1 binding to various RNAs is shown.

We agreed that the data of full-length hnRNP A2/B1 binding to the RNAs with or without m6A modification are important to support our conclusion. Therefore, in our previous manuscript, we showed the EMSA data that the full-length hnRNP A2/B1 binds the unmethylated m6A core sequence 5'-GGACU-3' slightly better than the methylated RNA (Fig. 6e and supplementary Figure 7b). We also tried to cleave the Sumo tag, but the full-length hnRNP A2/B1 without fusion tag is not stable and mostly degraded (Supplementary Figure 7c), while the tendency of ITC data (Supplementary Figure 7d) is similar to EMSA that the full-length hnRNP A2/B1 binds unmethylated RNA slightly better. To answer if the RGG box in the prion-like domain contributes to the RNA binding, we also purified a construct (residue 1-249) containing RRM domains and RGG box (Supplementary Figure 7e and f), and did ITC experiments with two different RNAs, 8mer and 10mer with/without m6A modification. The results showed that the hnRNP A2/B1 (1-249) binds RNAs with a tendency similar to the full-length protein and RRM domain protein (Supplementary Figure 7g).

Reviewer #2 (Remarks to the Author):

hnRNP A2/B1 is a RNA-binding protein involved with mRNA splicing, processing, export, and primary microRNA processing. It forms complexes with at least 20 other different hnRNPs and heterogeneous nuclear RNAs in the nucleus. The authors solved high resolution crystal structures of hnRNP A2/B1 with various RNA substrates, illustrating the molecular mechanism underline AGG and UAG motif recognition by A2B1 RRM1 and RRM2 domains. Together with biochemical and

bioinformatics analysis, they demonstrate hnRNP A2/B1 works as a RNA matchmaker by binding different RNA substrates, rather than a direct m6A reader. The crystal structures provide direct and solid evidence for the RNA substrate recognition by hnRNP A2/B1 and the in vitro EMSA and ITC results further confirm the conclusion. With YTHDC1 as a positive control, the authors employ bioinformatic analysis to demonstrate that only limited m6A-seq clusters overlap with the hnRNP A2/B1 tag clusters and there is no correlation between m6A intensity and hnRNP A2/B1 binding, suggesting that hnRNP A2/B1 might play a role in m6A structural switch. The multivalent binding characteristics of the hnRNP A2/B1 protein and the antiparallel orientation of the RNA strands bound by two RRM domains provide the molecular basis for hnRNP A2/B1-mediated RNA granule formation, which is significant insights given importance of this protein. I support publication of this work.

Response:

We thank the reviewer for his very positive comments on our study.

Reviewer #3 (Remarks to the Author):

The manuscript by Wu et al reports a comprehensive structural and biochemical analysis of RNA recognition by hnRNP A2/B1. The author determined the first crystal structures of the tandem RRM domains of hnRNP A2/B1 bound to five different RNAs. These structures combined with binding data reveal the conserved and variable features of RNA recognition for two RRM domains. The authors show that RRM1 and RRM2 have different specificity for AGG and UAG, respectively, and different stringency in recognition. Although tens of RRM-RNA complex structures are available, the RNA binding specificity of each RRM is determined by intricate and unpredictable interactions and has to be characterized individually. One important feature of these structures is the antiparallel arrangement of two RRM domains, which is different from most other tandem RRM-RNA structures. This arrangement has important consequence in RNA recognition and requires the two target RNAs recognized by RRM1 and RRM2 to be folded back or from different molecules.

hnRNP A2/B1 has been implicated in recognizing m6A modification. The current study provides evidence that hnRNP A2/B1 does not directly recognize m6A, but disfavors m6A, clarifying its role in m6A recognition.

Overall, the study is technically sound and the structural data are of high quality. The study reveals the structural basis of binding specificity of hnRNP A2/B1 and the unusual arrangement of RRM domains and is important for understanding the in vivo binding targets and function of hnRNP A2/B1.

Major concerns:

1) In all the determined structures, the two RRM domains adopt the same unusual antiparallel arrangement. It is probably the reason why the RNA strands in all the crystals are bound to different protein molecules, because the linker between two target sites is too short to allow the RNA adopt a fold-back conformation. The antiparallel arrangement and the interactions between two RRM domains is an important feature of hnRNP A2/B1 structure. My major concern is whether the antiparallel arrangement is caused by crystal packing? Could the authors show whether the RRMs domains are fixed in solution. Is there any interaction between two RRMs domains when they are separated?

Response:

We thank the reviewer for his positive comments on our study and insight on the antiparallel arrangement of two RRM domains. We believe that the antiparallel arrangement of two RRM domains in our structures is not caused by the crystal packing. All five crystal structures in this study have different space groups or/and cell dimensions, suggesting that the crystal packing is different in these structures. Similar arrangement is also observed in the crystal structures of two RRM domains of hnRNP A1 (also called UP1) (Rui-Ming Xu *et al.*, *Structure*, 1997; Jacqueline Vitali *et al.*, *Nucleic Acid Research*, 2002). Superimposition of our structures with hnRNP A1 indicates that the interactions between two RRM domains in both hnRNP A2/B1 and hnRNP A1 are similar (Supplementary Figure 5b), which fix the antiparallel arrangement.

We do not have the solution structure of hnRNP A2/B1, but the NMR structure of hnRNP A1 in solution indicated that two RRM domains have same antiparallel arrangement as in crystal structures (Supplementary Figure 5a). Due to similar interactions between two RRM domains in both hnRNP A2/B1 and hnRNP A1, we believe that the antiparallel arrangement of two RRM domains in hnRNP A2/B1 is also fixed in solution.

To answer the question if there is any interaction between two RRM domains when they are separated, we purified the RRM1 and RRM2 domains separately, and run the size exclusive column Superdex 75 10/30 (GE) after mixing the proteins of RRM1 and RRM2 domains with 1:1 ratio. The result indicated that the RRM1 and RRM2 domains do not form a stable complex in the gel-filtration condition (Supplementary Figure 5c), maybe because the interaction between RRM1 and RRM2 domains in solution when they are separated as two proteins is much weaker than when they are connected as one protein.

Reference:

1. Rui-Ming Xu *et al.* (1997). Crystal structure of human UP1, the domain of hnRNP A1 that contains two RNA-recognition motifs. *Structure*, 5(4), 559-570.
2. Vitali, J. *et al.* (2002). Correlated alternative side chain conformations in the RNA-recognition motif of heterogeneous nuclear ribonucleoprotein A1. *Nucleic Acids Research*, 30(7), 1531-1538.

Minor concerns:

1) *The protein-RNA interaction is described in too much detail. As the interactions of RRM1 are similar in the 8-mer and 10-mer complexes, repetitive description should be avoided.*

Response:

We deleted repetitive description for AGG recognition by the RRM1 in 10-mer complex, and keep some description for more specific recognitions of A1 and G2 observed in 10-mer complex due to the higher resolution.

REVIEWERS' COMMENTS:

Reviewer #1 (Remarks to the Author):

The reviewers have addressed my prior concerns and I recommend publication.

Reviewer #3 (Remarks to the Author):

I am satisfied with the authors' answers to my questions. I support publication of this paper.